

# Observations on hygroscopic growth and phase transitions of mixed 1, 2, 6-hexanetriol/(NH₄)₂SO₄ particles: Investigation of liquid-liquid phase separation (LLPS) dynamic process and mechanism and secondary LLPS

Shuai-Shuai Ma, Zhe Chen, Shu-Feng Pang, Yun-Hong Zhang

The Institute of Chemical Physics, School of Chemistry and Chemical Engineering, Beijing Institute of Technology, Beijing, 100081, China

*Correspondence to*: Shufeng Pang, Yunhong Zhang (sfpang@bit.edu.cn, yhz@bit.edu.cn)

**Abstract.** Atmospheric aerosols consisting of organic and inorganic components may undergo liquid-liquid phase separation
(LLPS) and liquid-solid phase transitions during ambient relative humidity (RH) fluctuation. However, the knowledge of dynamic phase evolution processes for mixed organic-inorganic particles is scarce. Here we present a universal and visualized observation on LLPS, efflorescence and deliquescence transitions as well as hygroscopic growth of mixed 1, 2, 6-hexanetriol/ammonium sulfate (AS) particles with different organic-inorganic mole ratios (OIR = 1:4, 1:2, 1:1, 2:1 and 4:1) with the high time resolution (0.5 s), using an optical microscope with a video camera. The optical images suggest that an
inner AS solution phase is surrounded by an outer organic-rich phase after LLPS for all mixed particles. The LLPS mechanism for particles with different OIRs differs, meanwhile, multiple mechanisms may dominate successively in individual particles with a certain OIR, somewhat inconsistent with earlier observations by literature. More importantly, another phase separation in inner AS solution phase, defined as secondary LLPS here, is observed for OIR = 1:1, 1:2 and 1:4 particles. The secondary LLPS may be attributed to the formation of more concentrated AS inclusions in the inner phase, and
becomes more obvious with decreasing RH and increasing AS mole fraction. Furthermore, the changes in size and amount of AS inclusions during LLPS are quantitatively characterized, which further illustrate the equilibrium partitioning process of organic and inorganic components. The experimental results have significant implications for revelation of complex phase transitions of internally mixed atmospheric particles and evaluation of liquid-liquid and liquid-solid equilibria in thermodynamic models.

**1 Introduction**

Atmospheric aerosols can undergo hygroscopic growth and phase transitions such as LLPS, efflorescence and deliquescence with ambient RH changing (Martin, 2000; Zuend et al., 2010; Shiraiwa et al., 2013), which dominate the size, physical state and morphology of particles, further causing a significant effect on scattering and absorption of solar light (Haywood and Boucher, 2000; Yu et al., 2005; Martin et al., 2004), gas-particle partitioning of semivolatile organics (Zuend et al., 2010;



Shiraiwa et al., 2013; Krieger et al., 2012), atmospheric heterogeneous chemistry such as $N_2O_5$ hydrolysis (Cosman et al., 2008; Thornton and Abbatt, 2005), and non-ideal mixing in $PM_{2.5}$ (Shiraiwa et al., 2013).

Field and laboratory studies showed that atmospheric particulate matters far away from local sources is basically the internal mixtures of organic and inorganic species (Middlebrook et al., 1998; Murphy et al., 2006; Lee et al., 2002), which is established by gas phase diffusion (Marcolli et al., 2004) and gas-particle partitioning (Zuend et al., 2010) of semivolatile

organic compounds. Organic species can dominate the fine aerosol mass with a mass fraction of 20-50% at continental mid-latitudes (Kanakidou et al., 2005). Sulfate (10-67%) neutralized by ammonium (6.9-19%) can also be measured in various regions (Zhang et al., 2007). Thus, mixed organic-sulfate particles can be regarded as model systems for troposphere aerosols to explore their hygroscopic growth and phase transitions.

Non-ideal thermodynamic behavior between organic and inorganic components in internally mixed particles can induce

LLPS into a mainly polar electrolyte-rich phase and a less polar organic-rich phase at phase separation relative humidity (SRH) (Erdakos and Pankow, 2004; Marcolli and Krieger, 2006). LLPS plays a significant role on morphology, chemical compositions, non-ideal mixing and gas-particle partitioning of atmospheric aerosols (Shiraiwa et al., 2013), as well as water uptake kinetics (Marcolli and Krieger, 2006; Hodas et al., 2016). Thus far, numerous studies have explored the LLPS that occurs in mixed particles consisting of various organic and inorganic species (Song et al., 2012b; Ciobanu et al., 2009; Zhou

et al., 2014; Bertram et al., 2011; O'Brien et al., 2015; Zuend and Seinfeld, 2012; Song et al., 2012a; Qiu and Molinero, 2015). As reported by literature, the oxygen to carbon elemental ratio (O:C) of organic species in the atmosphere is in the range of ~0.2 to ~1.0 (Ng et al., 2010; Heald et al., 2010; Zhang et al., 2007). Bertram et al. (2011) have found that LLPS in mixed sulfate-organic particles commonly occurred when the O:C < 0.7, and in some cases might be affected by the mass ratio of organics and sulfate. While for O:C > 0.7, no LLPS was observed. Song et al. (2012a) investigated the LLPS for a

series of model systems containing up to ten organic compounds mixed with sulfate and water. They found that LLPS always occurred in the mixtures with O:C < 0.56 and never occurred for O:C > 0.80; when 0.56 < O:C < 0.80, the occurrence of LLPS depended on the types and compositions of organic functional groups. Hence, the O:C ratio is proved to be an accurate predictor for the presence of LLPS, because the O:C represents the polarity of organic components and their miscibility with inorganics and water (Song et al., 2012a). Furthermore, Ciobanu et al. (2009) introduced three different

mechanisms for LLPS within PEG-400/AS/$H_2O$ particles depending on OIRs, i. e., nucleation-and-growth (OIR = 8:1 to 2:1), spinodal decomposition (OIR = 1.5:1 to 1:1.5) and growth of a second phase from the particle surface (OIR = 1:2 to 1:8). The spinodal decomposition occurs barrier-free in contrast to nucleation-and-growth, which has to overcome an energy barrier (Shelby, 1997; Papon et al., 1999). For nucleation-and-growth, subcritical nuclei are formed randomly within the liquid medium and begin to grow continuously once the critical size is attained (Ciobanu et al., 2009). Similarly, Song et al.

(2012b) found that LLPS for the C7 dicarboxylic acids/AS/$H_2O$ particles occurred by nucleation-and-growth, spinodal decomposition and growth of a second phase from the particle surface when sulfate dry mass fractions are < 0.30, 0.30 to 0.60 and 0.6 to 1.0, respectively. However, few studies focused on the equilibrium partitioning process of organic and inorganic components during LLPS.





In this work, we set up a high time resolution observation on hygroscopic growth and phase transitions of mixed 1, 2,
6-hexanetriol/(NH$_4$)$_2$SO$_4$ particles with different OIRs (1:4, 1:2, 1:1, 2:1 and 4:1) using a optical microscope with a video
camera. The optical images are captured with a time resolution of 0.5 s to determine the dynamic phase transition processes
and measure hygroscopic growth factors (GFs) during a RH cycle. The aims of this work are to: (1) provide an insight into
LLPS dynamic process of mixed organic-inorganic particles; (2) quantitatively characterize different LLPS mechanisms for
particles with different OIRs; (3) investigate the effect of organics on hygroscopic behaviours of sulfates for LLPS systems;
(4) explore the morphological changes and phase evolution processes of mixed particles during a RH cycle.

## 2 Experimental Section

### 2.1 Sample preparation

Five mixture solutions with different OIRs (1:4, 1:2, 1:1, 2:1 and 4:1) were prepared by dissolving 1, 2, 6-hexanetriol (99.0%
purity) and AS (99.0% purity) into ultrapure water (18.2 MΩ cm resistivity). The mixed solutions were aspirated and then
discharged by a syringe. Residual solutions in the syringe were pushed rapidly to spray the aerosol droplets onto
polytetrafluoroethylene (PTFE) substrates fixed in the bottom of the sample cells in two experimental systems.

### 2.2 Microscopic observations of single particles

The microscopic observations of single particles were performed by an optical microscope (Nikon Ti-S, 60×objective, 1.0
numerical aperture) coupled with a video camera. A similar experiment setup has been described detailed elsewhere (Ahn et
al., 2010; Song et al., 2012b), and thus a brief description was presented here. A ~14.13 cm$^3$ sample cell was fixed above an
inverted video microscopy. The PTFE substrate with deposited droplets was placed on a transparent glass slice in the bottom
of the sample cell. Another glass slice was fixed on the top of the sample cell to seal it. Mixed dry/wet N$_2$ streams with
changing water saturation ratios were passed through the sample cell to adjust the ambient RH. The total flow rate of N$_2$
streams is set up to ~900 sccm to rapidly reach equilibrium between ambient RH in sample cell and nitrogen flow RH. A
hygrometer (Centertek Center 313) was equipped at the outlet of sample cell to monitor the RH with an accuracy of ± 2.5%.
The RH was changed continuously at an average rate of 0.06-0.07% RH s$^{-1}$ in the RH range of ~10-90%. The optical images
of monitored particles were recorded every 0.5 s with a frequency of 2 frames s$^{-1}$. The size of particles with different OIRs
ranged from 55 μm to 80 μm at ~90% RH. All the measurements were performed at room temperature of 298 ± 1 K.

### 2.3 Raman measurements of single particles

The Raman measurements of mixed particles were achieved by using a Renishaw InVia confocal Raman spectrometer with a
Leica DMLM microscope (50×objective, 0.75 numerical aperture), which has been described detailed in previous studies
(Wang et al., 2005; Wang et al., 2017; Zhou et al., 2014). Briefly, a 514.5 nm laser and a 1800 g mm$^{-1}$ grating were adopted
to acquire the spectra in the range of 200-4000 cm$^{-1}$ with a resolution of 1 cm$^{-1}$. As mentioned above, the aerosol droplets



were sprayed onto the PTFE substrate in the bottom of the sample cell. Then, the sample cell was sealed by a transparent

polyethylene film. Mixed dry/wet $N_2$ streams were used to adjust the ambient RH, which was monitored by a hygrometer

(Centertek Center 313). Specially, the RH was changed stepwise in view of the accumulation time of 30 s for each spectral

measurement.

**2.4 Determinations of hygroscopic GFs**

The recorded images contained the information of morphology and size of particles. The changing particle sizes were

determined using an image analysing software (ToupView X64) with a fixed pixel and size ratio. The GFs could be

determined as

$$GF_{RH} = \frac{D_{RH}}{D_0} \tag{1}$$

where $D_{RH}$ was the diameter of mixed particles at a given RH and $D_0$ was the diameter of effloresced particles at < 10%

RH.

**3 Results and Discussions**

**3.1 Hygroscopic growth and phase transitions of mixed particles with OIR = 1:1**

**3.1.1 Hygroscopic growth of the OIR = 1:1 particle**

Fig. 1 shows the changes in GFs and morphology of a ~80 μm droplet (at ~90% RH) consisting of 1, 2, 6-hexanetriol and AS

with OIR = 1:1 during a RH cycle. For the dehumidification process, the droplet is first exposed to a high RH of ~91.1%

with a GF of ~1.50 at the beginning (time $t$ = 0). The particle remains as one single liquid phase, as shown in Fig. 1. As RH

decreases, LLPS occurs at ~79.3% RH, detected by the sudden appearance of schlieren (small separated regions), which will

be discussed detailed below. After that, two liquid phases are gradually formed, i. e., an inner AS solution phase and an outer

organic-rich phase. The water release continues with RH decreasing, showing a continuous reduction in GFs. The core-shell

particle undergoes a crystallization transition from ~47.7% RH to ~47.2% RH. However, the particle size continues to

decrease due to the continuous water release by the outer organic-rich phase consisting mainly of aqueous 1, 2, 6-hexanetriol,

which can absorb and release water continuously without any phase transition during the whole RH cycle, as shown in Fig.

S1 in the Supplementary Material. Thus, we can conclude that the nucleation of all the mixed droplets is owing to the

crystallization of AS. At very low RH, the smooth surface of the particle turns into irregular. Upon hydration, the outer phase

begins to take up water even at very low RH, showing the GFs increase slowly, similar to the hygroscopic behavior of pure 1,

2, 6-hexanetriol. At ~80.2% RH, the GFs begin to increase rapidly, meaning the dissolution of inner AS crystal, which can

also be identified by the images. Indeed, determination of deliquescence relative humidity (DRH) is always prone to

uncertainties, not like the SRH and efflorescence relative humidity (ERH), because the occurrence of AS crystal dissolution

is not easy to be clearly judged from the images and the turning point of humidification curve, in view of the continuous



water uptake by organic coating. At ~83.4% RH, the particle has been completely deliquesced, showing only one liquid
phase. Above this RH, the GFs are slightly lower than those in the dehumidification process, which should agree with the
weak volatility of 1, 2, 6-hexanetriol (Lv et al., 2019).

In addition, we determine the SRH, ERH and DRH of OIR = 1:1 particles with different particle diameters of 25-87 μm
(~90% RH), as shown in Fig. S2. The results show that, the SRH does not depend on particle sizes; the DRH shows no
dependency due to the large particle sizes (Gao et al., 2007; Ebert et al., 2002), considering that the DRH of particles smaller
than 60 nm increases with decreasing particle sizes (Hämeri et al., 2001; Russell and Ming, 2002); while the ERH decreases
slightly with the particle sizes, consistent with classical nucleation theory and earlier studies (Pant et al., 2004; Gao et al.,
2006).

### 3.1.2 Phase transition observations for the OIR = 1:1 particle

The optical images and corresponding illustrations for the same OIR = 1:1 particle during LLPS, secondary LLPS,
efflorescence and deliquescence are depicted in Fig. 2. During LLPS, first, the particle exists in homogenous mixed phase at
~79.4% RH, as shown in the first frame of Fig. 2. Note that both the bright globe in the center and the dark ring at the edge
are owing to the optical effect of light scattering (Bertram et al., 2011). When the RH arrives at ~79.3%, the schlieren over
the whole droplet appears suddenly, indicating the onset of LLPS by spinodal decomposition. It is noteworthy that the LLPS
mechanism can also be judged from the temporal changes of number of AS inclusions (Ciobanu et al., 2009; Song et al.,
2012b), which will be discussed detailed in Sec. 3.4. Then, the dispersed clusters grow and coalesce, leading to the separated
inclusions consisting mainly of AS solution, followed by the coalescence of these inclusions to form an inner AS solution
phase at $t = 275.0$ s. The large aggregation coexists with amounts of small AS inclusions, meaning the phase evolution
continuing till to an equilibrium (Ciobanu et al., 2009). From $t = 275.0$ s to 307.0 s, the AS inclusions become bigger and
merge into uniform AS solution phase. Till to ~75.2% RH, the equilibrium partitioning is reached, and the morphology
containing an inner AS solution phase linked to several inclusions and outer organic-rich shell is presented. Specially, a
secondary LLPS occurs at ~68.6% RH, showing a brighter aqueous phase present in the center of inner phase. The new
phase is attributed to more concentrated AS inclusions, as confirmed by our Raman spectra in next section, and becomes
more visible with decreasing RH. This is because, first, the inner AS-rich phase contains small amounts of organics; then,
the continuous water release would cause a gradual increase in sulfate concentration in the inner phase, which ultimately
results in the occurrence of secondary LLPS. For clarity, the concentrated AS inclusions are marked with different shades of
aqua in illustrations to indicate the degree of secondary LLPS. At 61.9% RH, the central AS inclusions can be clearly
distinguished. At $t = 735.0$ s, an AS crystal appears at the edge of the droplet, indicating the onset of efflorescence at 47.7%
RH. The following crystallization of AS phase and inclusions proceeds until 47.2% RH. Upon hydration, the solid AS
crystals begin to dissolve at the DRH of ~80.2%, and are deliquesced completely at ~83.4% RH. Note that the effloresced
particle transfers into homogenous mixed phase without LLPS after deliquescence, because the DRH of AS crystals is above
the SRH of the mixed particle.



### 3.1.3 Raman spectra analysis of the OIR = 1:1 particle

To clearly illustrate the LLPS and secondary LLPS of mixed particles, Raman spectra acquired on the surface and at the center of the OIR = 1:1 particle are collected at constant RH. The dehumidification process is shown in Fig. 3a and

humidification process for the effloresced particle is shown in Fig. 3b. Next to the spectra are the high-quality images corresponding to the same RH conditions from video microscopy, not the low resolution images by the Leica DMLM microscope (Fig. S3). As seen in Fig. 3, the O-H stretching vibration, $v$(O-H), of liquid water is identified at 3170-3715 cm$^{-1}$. The bands at 980 and 975 cm$^{-1}$ are assigned to the symmetric stretching vibration of $SO_4^{2-}$, $v_s(SO_4^{2-})$, in solution and crystalline states, respectively. The band of C-H stretching vibration, $v$(C-H), of 1, 2, 6-hexanetriol is observed at 2798-2995

cm$^{-1}$. Based on these, the intensity ratios of $v_s(SO_4^{2-})$ band to $v$(C-H) band are determined and depicted in Fig. 3c to identify the component distribution of mixed particle. For the dehumidification process, first, the particle exists as only one liquid phase at ~85.3% RH, confirmed by almost identical intensity ratios of a1, a2 and a3. Then, two separated phases are presented at ~77.0% RH. It is clear that the intensity ratio at the center increase significantly, and the intensity ratio of a5 is much higher than that of a4 and a6, indicating the morphology of an AS solution phase surrounded by an organic-rich shell.

Note that the signatures of both sulfate and organics can be observed in spectra a4-a6, suggesting there are a small amount of AS and 1, 2, 6-hexanetriol present in organic-rich and sulfate-rich phases, respectively. As the RH decreases to 68.1% and 58.2%, the intensity ratio at the center increases to 6.03 and 8.12, respectively, about 2-4 factors higher than that of a5, demonstrating the formation of more concentrated AS inclusions in the inner phase. When the RH increases to 76.3% after the particle is fully effloresced, there are an AS crystalline phase in the center and an organic-rich phase in the shell, as

identified by the spectra b1-b3. Specially, the band intensity of C-H in b2 is much higher than that in a5, a7 and a8, implying the presence of 1, 2, 6-hexanetriol in the veins of AS crystal as discussed below. At 84.5% RH, the intensity ratios of b4, b5 and b6 are almost the same, indicating the full deliquescence of particle.

## 3.2 Hygroscopic growth and phase transitions of mixed particles with OIR = 1:2 and 1:4

### 3.2.1 Hygroscopic growth of OIR = 1:2 and 1:4 particles

Fig. 4a and 4b display the GF changes and morphological changes in a RH cycle for mixed 1, 2, 6-hexanetriol/AS particles with OIR = 1:2 and 1:4, respectively. For the particle with OIR = 1:2, LLPS occurs at ~73.4% RH, and ends at ~73.2% RH. After that, the GFs decrease gradually until 41.9% RH, at which a rapid reduction in GFs appears. Hence, the ERH is ~41.9%. Upon hydration, first, a gradually faster increase in GFs is apparent, together with the transition of particle morphology from irregular to spherical-like. The dissolution of the inner AS crystal begins at ~77.3% RH, judged mainly

from the optical images.

The hygroscopic growth of the OIR = 1:4 particle follows an entirely different route from the OIR = 1:1 and 1:2 particles. First, the GFs decrease in an approximately linear manner upon dehydration until ~46.4% RH. Then, the onset and end of efflorescence is in a very narrow RH range around 46.4%, meaning a faster crystal growth due to the weaker transfer



limitation of water molecules, in view of the thinner viscous organic-rich shell for OIR = 1:4 particle. Final, the particle size
remains constant until 84.0% RH, and then increases steeply to the initial size. However, the DRH of mixed particle is
~80.2%, as identified by the occurrence of AS crystal dissolution with almost unchanged particle size under ~80.2% RH
shown in the optical images. Coupled with the morphological changes upon crystallization (Fig. S4), we can conclude that
aqueous 1, 2, 6-hexanetriol will enter into the veins of the AS crystal and then is enclosed by a crystalline AS crust, namely,
the organics may be trapped within the AS crystalline phase after crystallization. Likewise, Sjogren et al. (2007) reported
that effloresced AS crystals could exist in the veins or even were involved into organics. Ciobanu et al. (2009) introduced
that liquid PEG-400 could be trapped by the solid AS for the PEG-400/AS particle with OIR = 1:2. Song et al. (2012b) found
that the outer organic-rich phase was sucked into cavities of the inner AS crystal due to capillary forces. Upon hydration, the
GFs after full deliquescence overlap with those in the dehumidification process, suggesting almost no volatilization of 1, 2,
6-hexanetriol due to the capture by AS crystal.

**3.2.2 Phase transition observations for OIR = 1:2 and 1:4 particles**

The dynamic processes of LLPS, secondary LLPS and efflorescence for the same OIR = 1:2 and 1:4 particles are shown in
Fig. 5. For the OIR = 1:2 particle, LLPS occurs at ~73.4% RH ($t$ = 211.0 s), showing the schlieren appears suddenly. The
growth and coalescence of these regions lead to the generation of AS inclusions. The inclusions further merge together and
fade away, followed by the growth of a second phase from the rim of particle starting at ~219.0 s. Hence, LLPS is observed
to occur by first spinodal decomposition and then growth of a second phase from the particle surface. Here, the AS solution
phases in different illustrations during LLPS are marked with different shades of blue to indicate the degree of LLPS. Indeed,
the growth of a second phase is not always clearly distinguished from the images due to the optical effect, but it is clearly
visible in the Movie S5 (i.e., the LLPS process of the OIR = 1:2 particle). A secondary LLPS occurs at ~69.0% RH. The
number of concentrated AS inclusions increases with decreasing RH, along with the coalescence and growth of inclusions.
At ~41.9% RH, the inner phase turns into crystalline AS phase. In Fig. 5b, it is clear that LLPS for the OIR = 1:4 particle
occurs by growth of a second phase from the particle surface over a wider RH range of 78.3-76.3%. Specially, the secondary
LLPS occurs at ~77.9% RH, almost the same RH as the appearance of LLPS. Then, the degree of secondary LLPS increases
gradually along with the water release from the inner AS solution phase. The size of concentrated AS-rich phase in the
center of the AS solution phase almost attains the size of inner phase at ~50.7% RH, owing to the very high sulfate fraction.
The crystallization occurs at 46.4% RH. After ~2.0 s, the particle morphology transfers into rough, meaning rapid crystal
growth and trapping of organics into the cavity of AS crystal. Then, the crystal growth continues, resulting in the formation
of an AS crust (Fig. S4).



### 3.3 Hygroscopic growth and phase transitions of mixed particles with OIR = 2:1 and 4:1

#### 3.3.1 Hygroscopic growth of OIR = 2:1 and 4:1 particles

The hygroscopic cycles for mixed particles with OIR = 2:1 and 4:1 are shown in Fig. 6. LLPS occurs at ~73.6% and ~76.6% RH, respectively. The efflorescence begins at ~43.2% and ~50.6% RH, respectively. For the humidification process, the OIR = 2:1 particle first absorbs water very slowly at low RH, then the size of particle increases rapidly above ~50% RH; whereas, for the OIR = 4:1 particle, an appreciable size increase occurs even at very low RH (~20% RH), followed by a considerable and continuous size growth as RH increases, indicating a more similar hygroscopic property to pure 1, 2, 6-hexanetriol. Then,

the DRH for the two particles are ~74.9% and ~75.5%, respectively, as identified by the occurrence of crystal dissolution from the images.

#### 3.3.2 Phase transition observations for OIR = 2:1 and 4:1 particles

Fig. 7 shows the LLPS, secondary LLPS and efflorescence processes of the OIR = 2:1 and 4:1 particles. For the OIR = 2:1 particle, LLPS occurs first by spinodal decomposition at ~73.6% RH. After that, another LLPS mechanism, nucleation-and-

growth, is exhibited at about 72.9-71.7% RH. The ultimate particle morphology from nucleation-and-growth commonly consists of spherical droplets of the minor phase, i. e., AS inclusions in this case, dispersed in the major phase, i. e., organic-rich phase (Ciobanu et al., 2009). Crystallization of dispersed AS inclusions starts at ~43.2% RH and ends at ~41.4% RH. For the OIR = 4:1 particle, it is clear that the critical AS solution nuclei appear at ~76.6% RH, which further grow and coalesce into bigger spherical inclusions dispersed in the organic-rich phase until ~72.8% RH. Crystallization transition of

this particle occurs over a RH range of 50.6-50.2%. Furthermore, LLPS dynamic processes are also observed under two faster RH changing conditions, i. e., 0.14 RH s$^{-1}$ and 2.40 RH s$^{-1}$ (Fig. S5). Apparently, the number of formed AS inclusions increases significantly with higher RH changing rates. This is owing to the greater kinetics limitation in viscous outer organic-rich phase derived from the faster water release, which inhibit the coalescence of AS inclusions, as discussed by Fard et al. (2017).

### 3.4 Analysis of phase transition RH and LLPS processes

A summary of DRH, SRH and ERH of mixed 1, 2, 6-hexanetriol/AS particles with five different OIRs is shown in Fig. S6. The DRH values are just below the theoretical DRH of pure AS. This is because the AS crystal blends with organics in term of veins, resulting in water to dissolve AS partly at RH below the DRH due to capillary forces (Sjogren et al., 2007). The SRH for particles with different OIRs is around 75.0%, showing no dependency on OIRs. Also, the measured SRH is

slightly higher than the measurement results (~71.0% RH) by Bertram et al. (2011). The ERH of mixed particles is around 45.0%, which is in the range of typical ERH for heterogeneous nucleation of AS, i. e., > 40% RH, as reported by our previous study (Ma et al., 2019). The AS crystallization is not inhibited by the organic coating, owing to the weak water



diffusion limitation, consistent with the results from Robinson et al. (2013). Overall, the ERH and DRH of mixed particles are independent of the mole fraction of organics due to the occurrence of LLPS, as discussed by Bertram et al. (2011).

To illustrate the LLPS dynamic process and mechanism for mixed organic-inorganic aerosols, the number of distinguishable AS inclusions, the radius ratio of the largest AS inclusion and the AS solution phase to the whole particle (defined as $r_1$ and $r_2$, respectively), and corresponding RH as a function of time during LLPS are depicted in Fig. 8. The occurrence of LLPS is set to $t = 0$. For the OIR = 1:1 particle (Fig. 8c), the AS inclusions appear at about $t = 1.0$ s with a number of ~177 and $r_1$ of ~0.081. Then, the number shows a significant decrease with time, along with the rapid increase in $r_1$ until $t = 4.0$ s,

suggesting the coalescence of AS inclusions. At $t = 6.0$ s, the largest AS inclusion with $r_1$ of ~0.428 disappears, followed by the formation of an inner AS solution phase with $r_2$ of ~0.736. The number and $r_1$ change randomly with time after $t = 4.0$ s and 6.0 s, respectively, implying that AS inclusions are formed continuously, accompanied by the coalescence and merging into the AS solution phase. The $r_2$ decreases gradually with time, indicating an equilibrium partitioning process between organic-rich and AS-rich phases. After $t = 90.0$ s, the number, $r_1$ and $r_2$ remain almost unchanged, suggesting an equilibrium

arrangement is reached. In Fig. 8a, temporal changes in the number, $r_1$ and $r_2$ for the OIR = 1:2 particle show a similar trend to that of OIR = 1:1 particle in the prior period of LLPS (between black and red dash lines), owing to the same mechanism (spinodal decomposition). The number decreases from ~134 to ~23 from 1.5 s to 7.5 s, meanwhile, the $r_1$ increases from ~0.040 to ~0.195. At $t = 8.0$ s, the AS solution phase is formed with a size ratio $r_2$ of ~0.937 by another mechanism, i. e., growth of a second phase from the particle surface. After that, the number decreases gradually until $t = 21.0$ s, at which AS

inclusions disappear. Meanwhile, the $r_2$ decreases gradually to ~0.804. For the OIR = 1:4 particle, there is only one AS inclusion present during LLPS because LLPS occurs mainly by growth of a second phase from the surface of the particle. The AS solution phase appears at $t = 30.0$ s, followed by the continuous decrease in the size ratio. For the OIR = 2:1 particle, the temporal changes in the prior period of LLPS are similar to that of OIR = 1:1 and 1:2 particles, owing to the same LLPS mechanism, i. e., spinodal decomposition. The AS solution phase appears at $t = 13.5$ s. Then $r_1$ increases gradually with time

from $t = 17.0$ s, suggesting the growth of AS inclusions due to nucleation-and-growth. The new AS inclusions merge into the AS solution phase, resulting in the continuous increase in $r_2$. For the OIR = 4:1 particle, the number and $r_1$ first increase with time due to the nucleation and growth of AS inclusions. At $t = 17.5$ s, the AS solution phase is formed with the $r_2$ of ~0.165. Then $r_2$ and $r_1$ increase gradually, similar to that of OIR = 2:1 particle.

## 4 Summary and conclusions

The hygroscopic growth and phase transitions including LLPS, efflorescence and deliquescence are observed for mixed 1, 2, 6-hexanetriol/AS particles with different OIRs by combining microscope technique and Raman spectra. After LLPS, the core-shell structure of an inner AS solution core surrounded by an outer organic-rich shell is formed. For OIR = 1:2 and 2:1 particles, two different types of LLPS mechanism dominate successively during LLPS. For OIR = 1:1, 1:2 and 1:4 particles, a secondary LLPS in the inner phase, as a result of more concentrated AS inclusions formation, is exhibited with water





release, and it becomes more obvious with decreasing RH and increasing sulfate fractions. These results demonstrate a more complicated organic-inorganic partitioning process during RH fluctuation for mixed particles. Meanwhile, a special particle morphology of organics trapped by an AS crystal crust is observed for OIR = 1:4 particle. Besides, the quantitative characterization of LLPS dynamic processes further clarifies the different LLPS mechanisms for particles with different OIRs.

The complicated phase changes of atmospheric aerosols remain largely unclear until now, though they have significant effects on radiative forcing and atmospheric chemistry. Brown et al. (2006) found that reactive uptake coefficient of $N_2O_5$ on the surface of atmospheric particles was decreased significantly by the presence of a large amount of organics among the field measurements, which is due to the transfer limitation of $N_2O_5$ molecules caused by the formation of organic coatings. This finding was further validated by numerous laboratory observations on heterogeneous chemistry of $N_2O_5$ (Cosman et al.,

2008; McNeill et al., 2006; Mcneill et al., 2007; Badger et al., 2006). Among these, Cosman et al. (2008) found that the inhibition in $N_2O_5$ uptake coefficient was influenced significantly by types of aqueous solution phase, which should be further investigated. Furthermore, the water uptake of organic-inorganic mixtures might be affected by morphology effects drived from complex phase changes (Marcolli and Krieger, 2006). LLPS might affect the bulk-to-surface partitioning of organics, resulting in a considerable impact on droplet surface tension and cloud condensation nuclei (CCN) activity (Hodas

et al., 2016). Also, the gas-particle partitioning predictions would be extremely incorrect assuming that LLPS was ignored (Zuend and Seinfeld, 2012). Accordingly, the phase transition behaviours of mixed organic-inorganic particles should be comprehensively investigated to gain insights into complicated physical and chemical properties of atmospheric aerosols and provide valuable data for model simulations of phase evolution processes and heterogeneous reaction kinetics of environmental particles.


*Data availability.* Data are available at http://doi.org/10.5281/zenodo.3958966 (Ma et al., 2020).

*Author contributions.* SSM and YHZ designed the experimental plan. SSM preformed the measurements. ZC helped with data analysis. SSM and SFP wrote the paper. All authors discussed and contributed to the manuscript.


*Competing interests.* The authors declare that they have no conflict of interest.

*Acknowledgements.* This work was supported by the National Natural Science Foundation of China (No. 41875144 and 91644101).



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





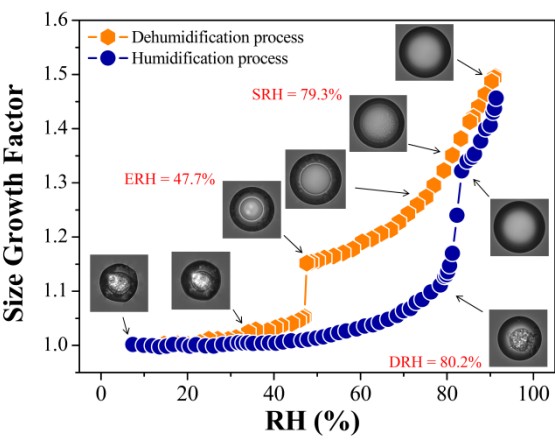

**Figure 1: Hygroscopic cycle of mixed 1, 2, 6-hexanetriol/AS particles with OIR = 1:1. The panels show the optical images corresponding to the GFs at the certain RH. The SRH, ERH and DRH values are given in red.**

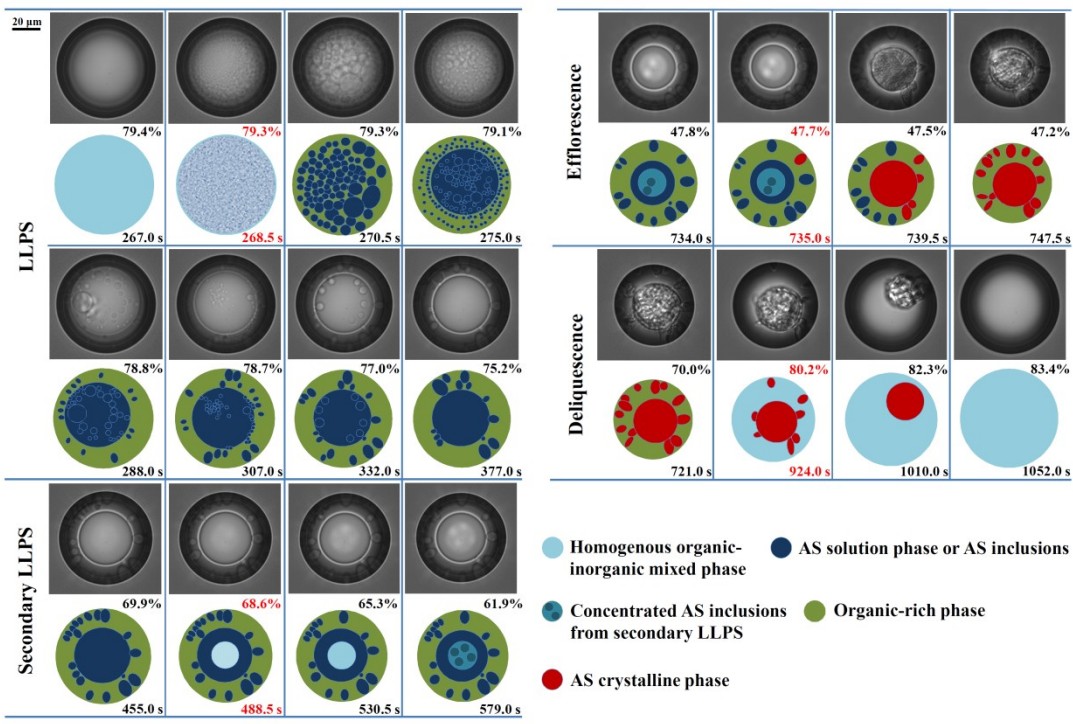

**Figure 2: Optical images and corresponding illustrations of mixed 1, 2, 6-hexanetriol/AS particle with OIR = 1:1 during LLPS, secondary LLPS, efflorescence and deliquescence. Below the optical images are the illustrations. The corresponding RH and time are given in each frame. The RH and time in red indicate the occurrence of phase transitions.**



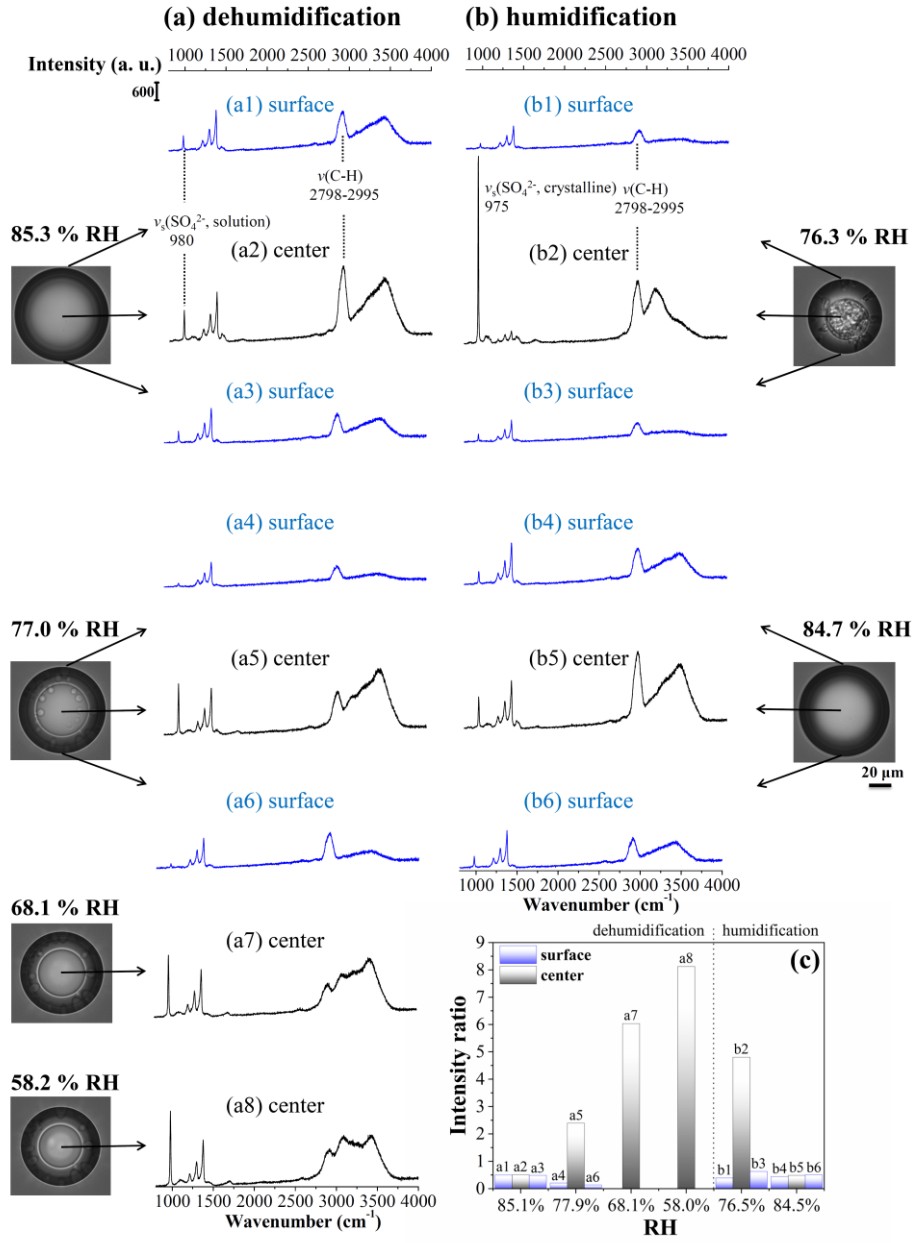

**Figure 3: Raman spectra acquired on the surface (blue) and at the center (black) of the OIR = 1:1 particle during dehumidification (a) and humidification (b), as well as intensity ratios of $v_s(SO_4^{2-})$ band to $v$(C-H) band among all the spectra (c). The corresponding microscopic images are shown near the spectra.**



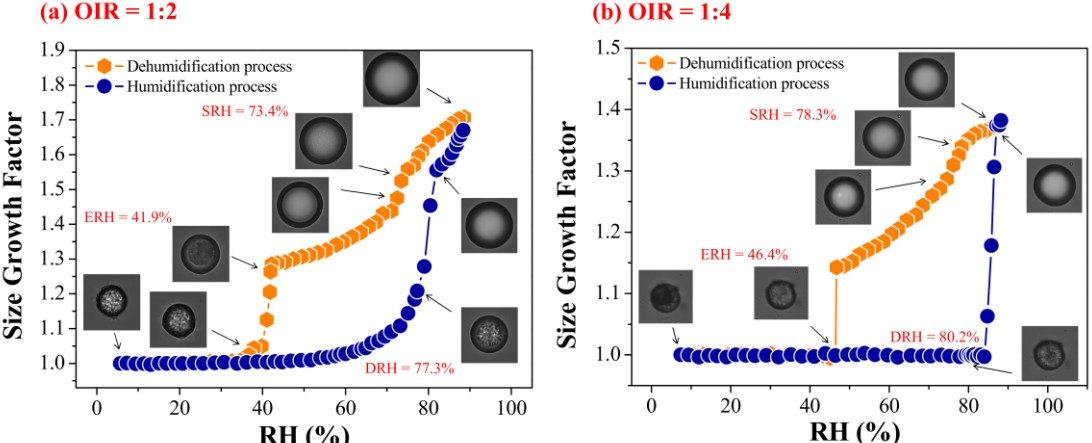

**Figure 4: Hygroscopic cycles of mixed 1, 2, 6-hexanetriol/AS particles with OIR = 1:2 (a) and 1:4 (b). The panels show the optical images corresponding to the GFs at the certain RH. The SRH, ERH and DRH values are given in red.**

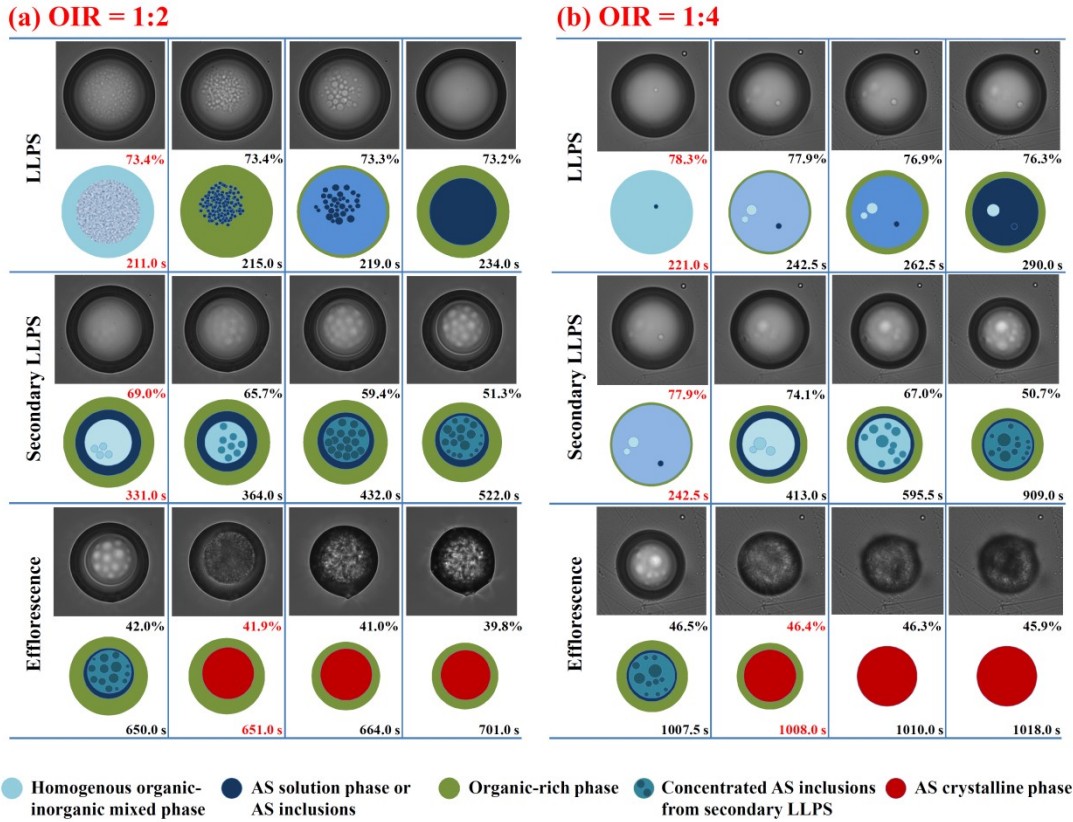

**Figure 5: Optical images and corresponding illustrations of mixed 1, 2, 6-hexanetriol/AS particles with OIR = 1:2 (a) and 1:4 (b) during LLPS, secondary LLPS and efflorescence. Below the optical images are the illustrations. The corresponding RH and time are given in each frame. The RH and time in red indicate the occurrence of phase transitions.**





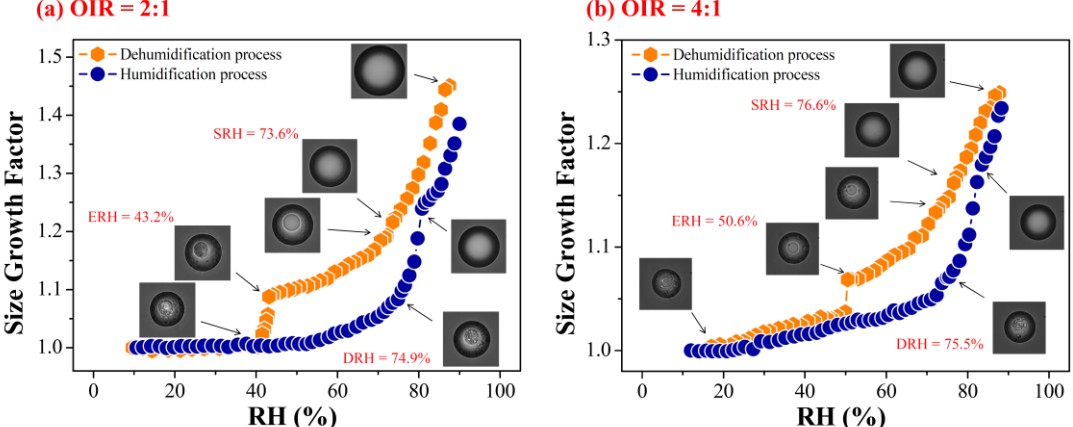

**Figure 6:** Hygroscopic cycles of mixed 1, 2, 6-hexanetriol/AS particles with OIR = 2:1 (a) and 4:1 (b). The panels show the optical images corresponding to the GFs at the certain RH. The SRH, ERH and DRH values are given in red.

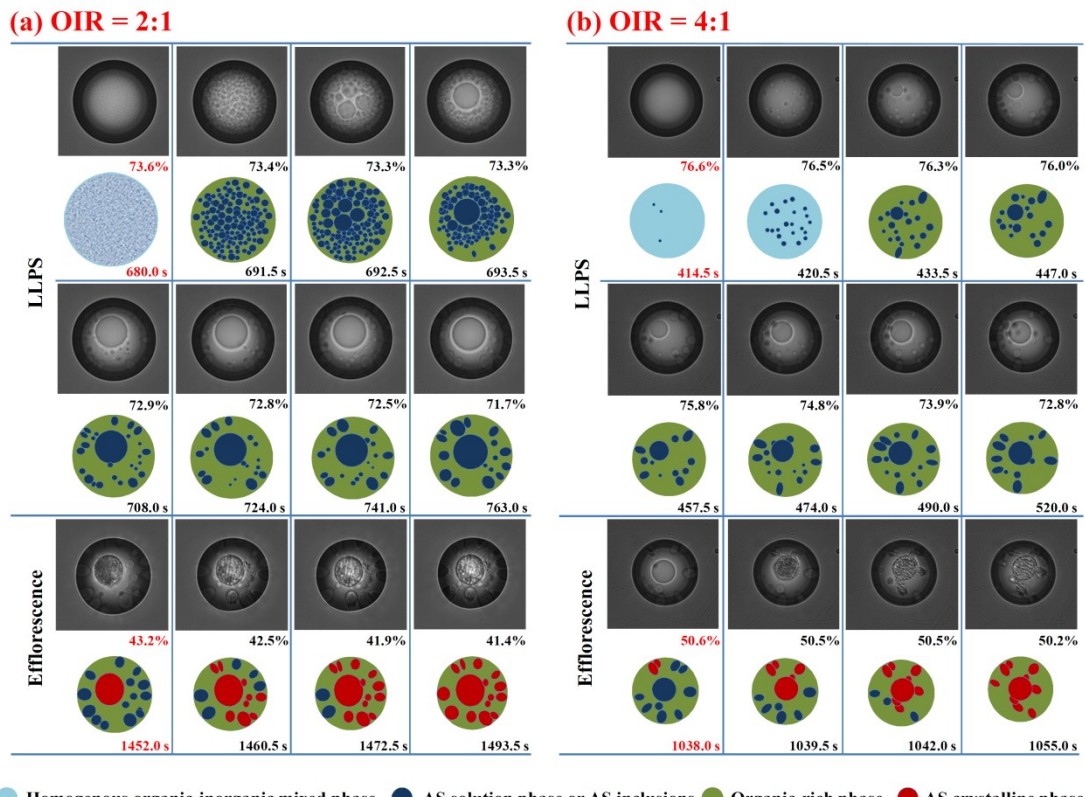

**Figure 7:** Optical images and corresponding illustrations of mixed 1, 2, 6-hexanetriol/AS particles with OIR = 2:1 (a) and 4:1 (b) during LLPS and efflorescence. Below the optical images are the illustrations. The corresponding RH and time are given in each frame. The RH and time in red indicate the occurrence of phase transitions.

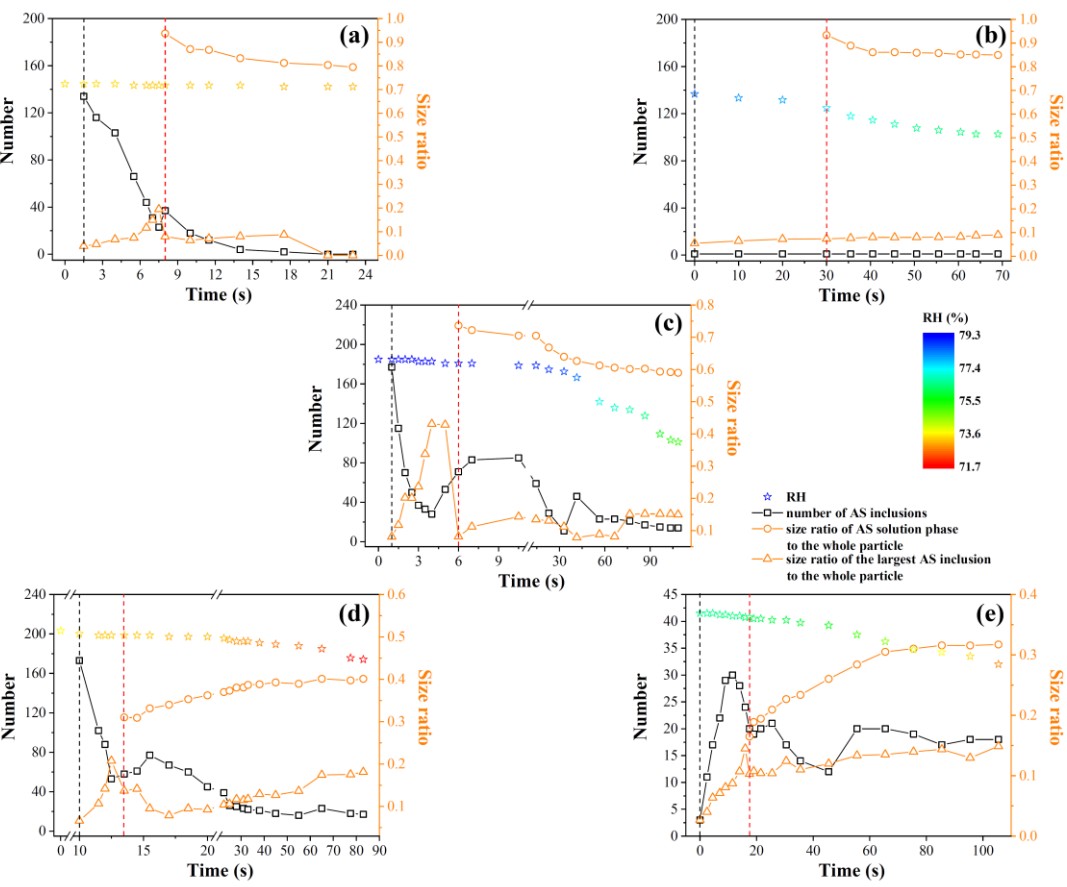


**Figure 8: Temporal changes in number of AS inclusions, size ratios of AS solution phases and the largest AS inclusions to the whole particles, as well as corresponding RH during LLPS for mixed 1, 2, 6-hexanetriol/AS particles with OIR = 1:2 (a), 1:4 (b), 1:1 (c), 2:1 (d) and 4:1 (e). The black dash line corresponds to the appearance of AS inclusions. The red dash line corresponds to the appearance of the AS solution phase.**
