# Peer review of "Observations on hygroscopic growth and phase transitions of mixed 1, 2, 6-hexanetriol/(NH4)2SO4 particles: Investigation of liquid-liquid phase separation (LLPS) dynamic process and mechanism and secondary LLPS during the dehumidification"

_Atmospheric Chemistry and Physics, 2020_

## Referee Comment (RC1)

This work investigated the hygroscopic growth and phase transitions for mixed particles composed of 1, 2, 6-hexanetriol and ammonium sulfate (AS) using an optical microscope and a Raman spectrometer. Liquid-liquid phase separation (LLPS) was observed in their measurements, and DRH, ERH, and SRH were determined for particles with different organic-inorganic molar ratios (OIR). Furthermore, a secondary LLPS phenomenon, confirmed by Raman spectra, was observed in their study, which is interesting and firstly explored. The manuscript is well-written and within the scope of this journal. I have several comments for consideration as below.

Comments:

Line 65: I suggest authors to give a brief introduction about the atmospheric significance of 1, 2, 6-hexanetriol, the organic species investigated in this work. Has it ever been detected in ambient aerosols? Or it was selected as a model species according to the O:C ratio, similar to Bertram et al., (2011)? This should be clarified in Introduction.

Line 100: Why not calculate GF using the image areas of particle at different RH and that of dry particle? How to estimate the diameter for the irregular or non-spherical particles in the software, especially for particles in effloresced state? The deviation between these two approximation methods should be estimated in this section. In addition, have the imaging pixel been calibrated in your measurement?

Line 149-150: "the continuous water release would cause a gradual increase in sulfate concentration in the inner phase, which ultimately results in the occurrence of secondary LLPS.". What cause the secondary LLPS? Why was it not observed in the study of Bertram et al. (2011)?

Line 178 and 179: I suggest to first introduce the result of mixed particles with OIR=1:4, followed by that of OIR=1:2. Also suggest for Line 200, Figure 4, 5 and 8 and related statements in main text, in an order with increasing molar fraction of organics.

Line 192-194: Please provide appropriate references to support the argument.

Line 240: When compared the phase transition RH with different OIR, I suggest to summarize the DRH, ERH and SRH values in a table in the revised manuscript for clear presentation. Of course, the DRH and ERH of AS, the results for the same systems investigated in previous study, i.e., Bertram et al. (2011), should also be

included for comparison.

Line 250 and Figure 8: Please clearly mention the temporal changes of LLPS dynamic process, not including secondary LLPS in the revised manuscript.

Figure 3(c): Please add the error bars.

---

## Community Comment (CC1)

This work investigated the hygroscopic growth and dynamic phase evolution processes of mixed 1, 2, 6-hexanetriol/AS particles. The dynamic LLPS processes and LLPS mechanisms for mixed particles with different organic-inorganic mole ratios were depicted quantitatively. More interestingly, the secondary LLPS was first explored when RH continued to decrease after LLPS. This work performs a comprehensive and detailed analysis on the complex LLPS behaviors of mixed organic-inorganic aerosols and provides insights into equilibrium partitioning processes of organic and inorganic components. Thus, I support the publication of this paper in ACP.

Comments:
1. Line 37: Mixed organic-sulfate particles can be regarded as model systems for troposphere aerosols? I think it is an inappropriate expression. The 1, 2, 6-hexanetriol/AS may be a model system for mixed aerosols that undergo LLPS, but it isn't a model system for atmospheric aerosols.
2. The authors suggested that the formation of brighter aqueous phase in the center of inner AS solution phase indicated the occurrence of secondary LLPS. How did the authors determine that this phenomenon come from secondary LLPS, not the optical effect?
3. In Fig. 3, b2 represented the intensity ratio of stretching vibration bands of crystalline $SO_4^{2-}$ to C-H, which differed with other intensity ratios, please indicate this in the caption.
4. Line 192: What does the "morphological changes upon crystallization" refer to? How can the authors conclude that aqueous 1, 2, 6-hexanetriol will enter into the veins of the AS crystal and then is enclosed by a crystalline AS crust?
5. Line 212: The secondary LLPS occurred at ~77.9% RH, almost the same RH as the appearance of LLPS, for the OIR = 1:4 particle with the highest sulfate fraction. Thus, whether the sulfate fraction can determine the RH at which the secondary LLPS occurs?
6. Line 245: The measured SRH in this work was slightly higher than the measurement results (~71.0% RH) by Bertram et al. (2011). Please discuss some of the sources of this discrepancy.
7. The sulfates are ubiquitous in atmospheric aerosols, but the concentration of 1, 2, 6-hexanetriol is limited. Why did the authors choose the 1, 2, 6-hexanetriol/AS as the research system?

---

## Author Response (AR1)

**Response to Reviewers:**

Thanks for the reviewer's comments on our manuscript entitled "Observations on hygroscopic growth and phase transitions of mixed 1, 2, 6-hexanetriol/(NH$_4$)$_2$SO$_4$ particles: Investigation of liquid-liquid phase separation (LLPS) dynamic process and mechanism and secondary LLPS".

The reviewers' comments are helpful for improving the quality of our work. The responses to the comments and the revisions in manuscript are given point-to-point below.

**Reviewer #1:**

1.   Line 37: Mixed organic-sulfate particles can be regarded as model systems for troposphere aerosols? I think it is an inappropriate expression. The 1, 2, 6-hexanetriol/AS may be a model system for mixed aerosols that undergo LLPS, but it isn't a model system for atmospheric aerosols.

**Author reply:** Thanks for the reviewer's suggestion. The sentence "Thus, mixed organic-sulfate particles can be regarded as model systems for troposphere aerosols to explore their hygroscopic growth and phase transitions." is revised to "Thus, mixed organic-AS

particles can be regarded as model organic-inorganic mixed systems and have been previously chosen in numerous laboratory studies." in lines 37 and 38 in the revised manuscript.

Meanwhile, the sentence "1, 2, 6-hexanetriol is chosen as a model organic species with O:C

< 0.7. Mixed 1, 2, 6-hexanetriol /(NH$_4$)$_2$SO$_4$ particles can be regarded as a model system for troposphere aerosols undergoing LLPS during the RH fluctuation." is added in lines 65-67 in the revised manuscript.

2.   The authors suggested that the formation of brighter aqueous phase in the center of inner AS

solution phase indicated the occurrence of secondary LLPS. How did the authors determine that this phenomenon come from secondary LLPS, not the optical effect?

**Author reply:** Thanks for the reviewer's suggestion. As mentioned in the text, the bright globe in the center and the dark ring at the edge were owing to the optical effect of light scattering when the particle existed in homogenous mixed phase. However, the optical effect could not cause any visual errors during the phase transitions, in other words, the dynamic processes of LLPS and secondary LLPS were not affected by the optical effect. Besides, the secondary LLPS processes could be clearly distinguished by the movies in the Supplement.

3.  In Fig. 3, b2 represented the intensity ratio of stretching vibration bands of crystalline $SO_4^{2-}$ to

C-H, which differed with other intensity ratios, please indicate this in the caption.

**Author reply:** Thanks for the reviewer's suggestion. We have added the sentence "Note that the value of b2 in (c) represents the intensity ratio of stretching vibration bands of crystalline

$SO_4^{2-}$ to C-H." in the caption of Fig. 3.

4.  Line 192: What does the "morphological changes upon crystallization" refer to? How can the authors conclude that aqueous 1, 2, 6-hexanetriol will enter into the veins of the AS crystal and then is enclosed by a crystalline AS crust?

**Author reply:** As shown in Fig. S4, the morphology of the OIR = 1:4 particle became more irregular and darker as the RH decreased. This indicated that the crystal growth continued at lower RH after efflorescence. Thus, we conclude that aqueous 1, 2, 6-hexanetriol enters into the veins of the AS crystal and then is enclosed by a crystalline AS crust. More importantly, the particle size remained constant until 84.0% RH, differing from the particles with the other

OIRs.

5.  Line 212: The secondary LLPS occurred at ~77.9% RH, almost the same RH as the appearance of LLPS, for the OIR = 1:4 particle with the highest sulfate fraction. Thus, whether the sulfate fraction can determine the RH at which the secondary LLPS occurs?

**Author reply:** The secondary LLPS occurred at ~68.6%, 69.0%, and 77.9% RH for OIR =

1:1, 1:2 and 1:4 particles, respectively. Moreover, no secondary LLPS was observed for OIR

= 2:1 and 4:1 particles. It is clear that the higher sulfate fraction tended to cause the occurrence of secondary LLPS, but the relationship between the sulfate fraction and the RH at which secondary LLPS occurs remains unclear and needs to be further investigated.

6.  Line 245: The measured SRH in this work was slightly higher than the measurement results (~71.0% RH) by Bertram et al. (2011). Please discuss some of the sources of this discrepancy.

**Author reply:** Indeed, the measured SRH values in our work were around 75.0%, slightly higher than the measurement results of 71.0% RH by Bertram et al. There are several possible sources of this discrepancy: First, the particle size employed in our work (55-80 μm) was larger than that in Bertram's work (10-30 μm); second, in both cases, the uncertainty in the measured SRH was 2.5% RH; finally, the rate of RH changes was 0.06-0.07% RH $s^{-1}$ in our work, higher than that of 0.4-0.6 % RH per minute in Bertram's work, probably causing higher measured SRH values in our work.

7.  The sulfates are ubiquitous in atmospheric aerosols, but the concentration of 1, 2,

6-hexanetriol is limited. Why did the authors choose the 1, 2, 6-hexanetriol/AS as the research system?

**Author reply:** In Bertram's work, LLPS in mixed sulfate-organic particles commonly occurred when the O:C < 0.7, while for O:C > 0.7, no LLPS was observed. Based on this, the

1, 2, 6-hexanetriol can act as a model organic matter with O:C < 0.7 and mixed 1, 2,

6-hexanetriol/AS represents a model system for mixed organic-inorganic aerosols which would undergo LLPS upon the RH fluctuation. Thus, we choose the 1, 2, 6-hexanetriol/AS

system to investigate the LLPS dynamic process and mechanism of mixed aerosols.

**Reviewer #2:**

1.  Line 65: I suggest authors to give a brief introduction about the atmospheric significance of 1,

2, 6-hexanetriol, the organic species investigated in this work. Has it ever been detected in ambient aerosols? Or it was selected as a model species according to the O:C ratio, similar to

Bertram et al., (2011)? This should be clarified in Introduction.

**Author reply:** Thanks for the reviewer's suggestion. The organic species in the mixed atmospheric aerosols consists of 1000s of different molecules, with only about 10% identified (Hallquist et al., 2009). In the present work, the 1, 2, 6-hexanetriol could act as a model organic species with O:C < 0.7. Thus, the 1, 2, 6-hexanetriol/AS mixed system represents a model system for mixed organic-inorganic aerosols undergoing LLPS. We have added the sentence "1, 2, 6-hexanetriol is chosen as a model organic species with O:C < 0.7. Mixed 1,

2, 6-hexanetriol /(NH$_4$)$_2$SO$_4$ particles can be regarded as a model system for troposphere aerosols undergoing LLPS during the RH fluctuation." in the Introduction.

2.  Line 100: Why not calculate GF using the image areas of particle at different RH and that of dry particle? How to estimate the diameter for the irregular or non-spherical particles in the software, especially for particles in effloresced state? The deviation between these two approximation methods should be estimated in this section. In addition, have the imaging pixel been calibrated in your measurement?

**Author reply:** Thanks for the reviewer's suggestion. The hygroscopic growth of aerosol particles is generally expressed as mass growth factors, i.e., the mass of particles at a given

RH divided by the mass of dry particles (Ma et al., 2019), or size growth factors, i.e., the diameter of particles at a given RH divided by the diameter of dry particles. In the present work, the size growth factors of mixed particles were determined by the optical images with an image analysing software to explore the hygroscopic behaviours of mixed particles, similar to the treatment of Sun et al. (2018). Indeed, the size of particles after efflorescence was estimated approximatively in the software, but the error of growth factors caused by such the approximation was negligible compared with the large GF values of homogeneous aqueous droplets. The ratio of image area of the OIR = 1:1 particle at different RH to that of the effloresced particle is determined by the same software, as shown in Fig. R1. It is clear that the image area ratio and the size growth factors show similar trends regarding the hygroscopic behaviours of the OIR = 1:1 particle. In addition, we have calibrated the imaging pixel with a fixed pixel and size ratio (1 μm = 10.667 pix).

[Figure]

Figure R1: Size growth factors (a) and imaging area ratio (b) of mixed 1, 2, 6-hexanetriol/AS particles with
OIR = 1:1 during the RH cycle.

3.  Line 149-150: "the continuous water release would cause a gradual increase in sulfate concentration in the inner phase, which ultimately results in the occurrence of secondary

LLPS.". What cause the secondary LLPS? Why was it not observed in the study of Bertram et al. (2011)?

**Author reply:** First of all, the phase separation in the mixed particles can be attributed to the salting out effect, i.e., the decrease in the solubility of organics in an aqueous salt solution. The correlation of the solubility of organics, $S$, and the concentration of the salt, $C_s$, can be expressed by the Setchenov equation (Lee, 1997): In $S/S_0 = k_sC_s$, where $S_0$ is the solubility of organics in water without the salt, $k_s$ is the Setchenov constant. Second, as confirmed by our Raman spectra in the present work, there were a small amount of AS and 1, 2, 6-hexanetriol present in organic-rich and sulfate-rich phases, respectively. Thus, as RH decreased after LLPS, the concentration of AS in the inner phase increased significantly and the solubility of organics decreased, resulting in the formation of more concentrated AS inclusions in the inner phase due to the salting out effect, similar to the occurrence of LLPS. In the study of Bertram et al. (2011), the size of observed particles was around 10-30 μm, while that was 55-80 μm in our work. We speculate that the secondary LLPS may not be clearly observed in the case of smaller particle size.

4. Line 178 and 179: I suggest to first introduce the result of mixed particles with OIR=1:4, followed by that of OIR=1:2. Also suggest for Line 200, Figure 4, 5 and 8 and related statements in main text, in an order with increasing molar fraction of organics.

**Author reply:** Thanks for the reviewer's suggestion. We have adopted reviewer's advice and revised our manuscript accordingly.

5. Line 192-194: Please provide appropriate references to support the argument.

**Author reply:** Thanks for the reviewer's suggestion. We have adopted the reviewer's advice in the revised manuscript.

6. Line 240: When compared the phase transition RH with different OIR, I suggest to summarize the DRH, ERH and SRH values in a table in the revised manuscript for clear presentation. Of course, the DRH and ERH of AS, the results for the same systems investigated in previous study, i.e., Bertram et al. (2011), should also be included for comparison.

**Author reply:** Thanks for the reviewer's suggestion. We have adopted the reviewer's advice. The Table S1 summarized the DRH, ERH and SRH of mixed particles with different OIRs has been placed in the Supplement.

7. Line 250 and Figure 8: Please clearly mention the temporal changes of LLPS dynamic process, not including secondary LLPS in the revised manuscript.

**Author reply:** Thanks for the reviewer's suggestion. We have adopted the reviewer's advice and added the sentence "not including secondary LLPS" in line 271 in the revised manuscript.

8.   Figure 3(c): Please add the error bars.

**Author reply:** Thanks for the reviewer's suggestion. We have adopted the reviewer's advice and revised our manuscript accordingly.

Reference:

Hallquist, M., Wenger, J. C., Baltensperger, U., Rudich, Y., Simpson, D., Claeys, M., Dommen, J.,
Donahue, N. M., George, C., Goldstein, A. H., Hamilton, J. F., Herrmann, H., Hoffmann, T., Iinuma, Y.,
Jang, M., Jenkin, M. E., Jimenez, J. L., Kiendler-Scharr, A., Maenhaut, W., McFiggans, G., Mentel, T.
F., Monod, A., Prévôt, A. S. H., Seinfeld, J. H., Surratt, J. D., Szmigielski, R., and Wildt, J.: The
formation, properties and impact of secondary organic aerosol: current and emerging issues, Atmos.
Chem. Phys., 9, 5155-5236, 2009.
Lee, L. L.: A molecular theory of Setchenov's salting-out principle and applications in mixed-solvent
electrolyte solutions, Fluid Phase Equilibr., 131, 67-82, 1997.
Ma, S. S., Yang, W., Zheng, C. M., Pang, S. F., and Zhang, Y. H.: Subsecond measurements on aerosols:
From hygroscopic growth factors to efflorescence kinetics, Atmos. Environ., 210, 177-185, 2019.
Sun, J. X., Liu, L., Xu, L., Wang, Y. Y., Wu, Z. J., Hu, M., Shi, Z. B., Li, Y. J., Zhang, X. Y., Chen, J.
M., and Li, W. J.: Key role of nitrate in phase transitions of urban particles: Implications of important
reactive surfaces for secondary aerosol formation, J. Geophys. Res.-Atmos., 123, 1234-1243, 2018.

**Reviewer #3:**

Major comment:

The authors claimed that more concentrated AS inclusions form inside the inner sulfate rich core, and this is defined as a secondary LLPS. Optical images indeed indicate heterogeneity in the inner core. The Raman spectra showing different sulfate/organic ratios in Fig 3, however, were actually taken from different RH values. I am not fully convinced that these Raman spectra indicate a secondary LLPS within the inner core. Do Raman spectra within the inner core measured at the same RH show different intensity ratios? What is the thermodynamics explanation of the proposed secondary LLPS? Is the secondary LLPS an equilibrium state or a transitional state?

**Author reply:** The Raman spectra acquired at the same position under the same RH have the almost identical intensity ratios. For the secondary LLPS, first, the LLPS in the mixed phase is related to the salting out effect, i.e., the decrease in the solubility of organics in an aqueous salt solution. The correlation of the solubility of organics, $S$, and the concentration of the salt, $C_s$, can be expressed by the Setchenov equation: In $S/S_0 = k_sC_s$, where $S_0$ is the solubility of organics in water without the salt, $k_s$ is the Setchenov constant. Second, there were a small amount of 1, 2, 6-hexanetriol present in the inner sulfate-rich phase. The concentration of AS in the inner phase would increase significantly with decreasing RH after LLPS, resulting in the formation of more concentrated AS inclusions in the inner phase due to the salting out effect. Finally, according to the phase rule, the degree of freedom of the mixed system is zero in the case of coexistence of the three liquid phases during secondary LLPS. In the phase diagram of three pairs of partially miscible systems, the concentration of the three phases cannot be changed when the three phases coexist, but the relative content of the three phases can be changed according to the position of the system points in the phase diagram. Thus, we conclude that the mixed system is in an equilibrium state during the secondary LLPS. The sentences "Based on the phase rule, the degree of freedom of the mixed system is zero in the case of coexistence of the three liquid phases during secondary LLPS. In the phase diagram of three pairs of partially miscible systems, the concentration of the three phases cannot be changed when the three phases coexist, but the relative content of the three phases can be changed according to the position of the system points in the phase diagram." have been added in lines 153-156 in the revised manuscript.

Minor comments:

Introduction: it might be worth to mention that LLPS can also occur for organic mixtures, such as in secondary organic aerosol (e.g., https://acp.copernicus.org/articles/16/7969/2016/ https://acp.copernicus.org/articles/17/11261/2017/;https://www.nature.com/articles/s41467-018 -06622-2 )

**Author reply:** Thanks for the reviewer's suggestion. We have adopted the reviewer's advice and revised our manuscript accordingly. The sentence "Recent studies also show that LLPS can occur in mixed organic systems without inorganic salts, causing more hydrophilic and less hydrophilic phases under high RH conditions (Renbaum-Wolff et al., 2016; Song et al., 2017; Liu et al., 2018)." is added in lines 41-43 in the revised manuscript.

state in the caption that LLPS and secondary LLPS are measured during the dehumidification

**Author reply:** Thanks for the reviewer's suggestion. We have adopted the reviewer's advice and revised the caption of the manuscript.

Page 4 Line 119-120: slowly -> gradually, rapidly-> abruptly. Changes of GF values with RH are thermodynamic processes, rather than kinetic processes.

**Author reply:** Thanks for the reviewer's suggestion. We have adopted the reviewer's advice and revised our manuscript accordingly. In line 121, the work "slowly" is revised to "gradually". In lines 122 and 226, the work "rapidly" is revised to "abruptly".